# Crimean-Congo hemorrhagic fever virus NSm protein inhibits the type I interferon signaling by binding to STAT2

Rokusuke Yoshikawa[1,2], Yasuteru Sakurai[1,2,3], Sayako Kondo[2], Mayuko Kimura[2], Jiro Yasuda [1,2,3,4]*

1 Department of Emerging Infectious Diseases, Institute of Tropical Medicine (NEKKEN), Nagasaki University, Nagasaki, Japan, 2 Department of Emerging Infectious Diseases, National Research Center for the Control and Prevention of Infectious Diseases (CCPID), Nagasaki University, Nagasaki, Japan, 3 School of Tropical Medicine and Global Health, Nagasaki University, Nagasaki, Japan, 4 Graduate School of Biomedical Sciences, Nagasaki University, Nagasaki, Japan

* j-yasuda@nagasaki-u.ac.jp

## Abstract

The type I interferon (IFN-I) response, which includes IFN-I induction and signaling, plays an important role in a host's defence against viral infections. Many pathogenic viruses target it to evade the host immunity. Crimean-Congo hemorrhagic fever virus (CCHFV), the causative agent of Crimean-Congo hemorrhagic fever, which features high mortality in humans, has been reported in southeastern Europe, Africa, the Middle East, and Asia. Although a previous study reported that CCHFV antagonizes IFN-I signaling in human cell lines, it is unclear how it inhibits IFN-I signaling. Here we demonstrated that the non-structural protein of CCHFV, NSm, suppresses IFN-I signaling in human cell lines. Furthermore, we discovered that NSm binds to STAT2, an important host protein in IFN-I signaling, and induces its degradation within cells. Taken together, our results imply that NSm suppresses IFN-I signaling by targeting human STAT2.

## Author summary

Crimean-Congo hemorrhagic fever virus (CCHFV), the causative agent of Crimean-Congo hemorrhagic fever, which features high mortality in humans. However, no effective vaccines and drugs are currently available to prevent CCHF. Although a previous study reported that CCHFV antagonizes IFN-I signaling in human cell lines, it is unclear how it inhibits IFN-I signaling. In this study, we revealed that the non-structural protein of CCHFV, NSm, inhibits IFN-I signaling in human cell lines. In addition, we indicate that the binding of NSm to STAT2 results in suppression of IFN-I signaling.

**Data availability statement:** All data are in the manuscript and its Supporting information files.

**Funding:** This research was supported by the Moonshot Research & Development program from the Japan Science and Technology Agency (JST) under Grant Number JPMJMS2025 (J.Y.), grants from the Japan Agency for Medical Research and Development (AMED) under Grant Numbers JP21fk0108080, JP22fk0108114, JP24fm0208101, and JP243fa627004 (J.Y.), the Japan Society for the Promotion of Science under Grant Number 21K08493, and 25K11741 (R.Y.), the Takeda Science Foundation (R.Y.), and the Naito Foundation (R. Y.). The funders had no role in study design, data collection and analysis, decision to publish, or preparation of the manuscript.

**Competing interests:** The authors have declared that no competing interests exist.

## Introduction

Crimean-Congo hemorrhagic fever (CCHF) is a tick-borne viral disease caused by the CCHF virus (CCHFV), an *Orthonairovirus* genus belonging to the order *Bunyavirales* and family *Nairoviridae*. CCHF is endemic to southeastern Europe, Africa, the Middle East, and Asia [1]. In humans, CCHF is characterized by severe hemorrhagic fever with a fatality rate of 10–40% [2]. However, the pathogenic mechanism of CCHFV remains poorly understood, and there are currently no effective vaccines against CCHFV or antivirals available for treating CCHF.

The CCHFV genome comprises three negative-stranded RNA segments (S, M, and L). The L segment encodes the viral RNA-dependent RNA polymerase. The S segment encodes the nucleoprotein (NP) and non-structural proteins (NSs), which are important for regulating the apoptosis of infected cells [3]. The M segment encodes a glycoprotein precursor (GP) that is cleaved by intracellular proteases such as furin into structural envelope glycoproteins (Gn and Gc) and the NSm non-structural protein after translation [4]. Gn and Gc play critical roles in the attachment and entry of viral particles into cells [4]; however, the function of NSm remains unclear.

The innate immune response induced by type I interferon (IFN-I) is important for preventing viral infections [5]. Antiviral innate immunity is initiated by the interaction of viral RNA with cellular pattern recognition receptors such as retinoic acid–inducible gene I–like receptors, transmembrane toll-like receptor 3, and melanoma differentiation associated gene 5 [6]. Upon recognition, the signaling cascade induces IFN-I expression (IFN-I induction phase). The attachment of secreted IFN to IFN receptors results in the phosphorylation of signal transducer and activator of transcription 1 (STAT1) and STAT2. Hetero- or homodimers of phosphorylated STATs form heterotrimeric IFN-stimulated gene factor 3 (ISGF3) with IFN regulatory factor-9. The binding of ISGF3 translocated into the cellar nucleus to an IFN-stimulated response element (ISRE) induces the expression of antiviral IFN-stimulated genes (ISGs) [7].

Welch *et al*. recently generated the NSm-deficient CCHFV (CCHFVΔNSm) by a reverse genetics system [8]. In A549 cells, which are IFN-I-competent, the growth rate of CCHFVΔNSm was lower than that of wild-type CCHFV [8]. In contrast, the growth speed and end point titres of CCHFVΔNSm and wild-type CCHFV are comparable in BSR-T7/5 cells, which are IFN-I-deficient [8]. In addition, CCHFV replication is reportedly not inhibited by subsequent IFN-I treatment [9]. Therefore, NSm may be involved in interrupting IFN-I signaling. Here we investigated whether NSm inhibits IFN-I signaling and further examined the mechanism of anti-IFN-I signaling by NSm.

## Materials and methods

### Cell culture

Human embryonic kidney (HEK) 293T (CRL-11268; ATCC) and Madin-Darby canine kidney (MDCK) (CCL-34; ATCC) cells were cultured in Dulbecco's modified Eagle's medium (Sigma-Aldrich) supplemented with 10% heat-inactivated fetal calf serum and 1% penicillin/streptomycin (Thermo Fisher Scientific).

## Plasmids

The open reading frame (ORF) sequence of the GP of the CCHFV Hoti strain was synthesized (Genewiz). The synthesized GP sequence was codon-optimized for high expression in human cells and inserted into pCAGGS (pCAGGS/CCHFV GP). The expression plasmid for CCHFV GPΔNSm was constructed by removal of the NSm coding region from pCAGGS/CCHFV GP using a KOD-Plus mutagenesis kit (Toyobo). The ORF encoding the NSm from the pCAGGS/CCHFV GP was amplified by polymerase chain reaction (PCR) and inserted into pCAGGS with a FLAG or His tag using an In-Fusion HD cloning kit (TaKaRa). Hemagglutinin (HA)-tagged Severe fever with thrombocytopenia virus (SFTSV) NSs, His-tagged STAT1, His-tagged STAT2, and FLAG-tagged STAT2 expression plasmids were prepared as described previously [10]. To construct the green fluorescent protein (GFP)–fused STAT1 or STAT2 expression plasmids, the ORF encoding STAT1 or STAT2 was inserted into a pAC-GFP N1 plasmid (Clontech). Expression plasmids for a series of STAT2 deletion mutants were constructed using a KOD-Plus mutagenesis kit (Toyobo). Expression plasmids for STAT1(2SH) and STAT2(1SH) were constructed using the In-Fusion HD cloning kit (TaKaRa). The expression plasmid for HA-tagged ubiquitin with only K48 and other lysines mutated to arginines, pRK5-HA-Ubiquitin-K48, was a gift from Ted Dawson (Addgene plasmid #17605; http://n2t.net/addgene:17605; RRID:Addgene_17605) [11].

## Virus

The vesicular stomatitis Indiana virus (VSIV) was a kind gift from Dr. H. Kida (Hokkaido University). Virus stocks were prepared from the culture supernatants of MDCK cells.

## Reporter gene assay

The reporter gene assay was performed as previously described [10,12]. Briefly, HEK293T cells ($2 \times 10^5$ cells/well in 12-well plates) were transfected with the ISRE reporter plasmid containing the ISRE promoter upstream of the firefly luciferase gene (500 ng; Promega), and the pRL-TK plasmid (Renilla luciferase control plasmid driven by the constitutively active herpes simplex virus thymidine kinase promoter; 50 ng; Promega) as an internal control. Transfections were performed using TransIT-LT1 transfection reagent (Mirus Bio, Cat# MIR 2300) according to the manufacturer's instructions, with or without the indicated amount of NSm-FLAG expression plasmid. At 24 h post-transfection, the cells were treated or untreated with IFN-αA/D (500 U/mL) (PBL Assay Science) for 18 h. Luciferase activities were then measured using a Dual-Luciferase Reporter Assay kit (Promega) and a TriStar LB941 system (Berthol). Firefly luciferase activity was normalized to Renilla luciferase activity to account for transfection efficiency.

## Co-localization assay

All transfections were performed using TransIT-LT1 transfection reagent (Mirus Bio, Cat# MIR 2300) according to the manufacturer's instructions. To investigate STAT1 and STAT2 localizations, the expression plasmids for His tagged STAT1 and STAT2 were transfected into HEK293T cells with pDsRed2-ER (Clontech) or pDsRed-Monomer-Golgi (Clontech) and the expression plasmid for NSm-FLAG. The cells were subjected to an indirect immunofluorescence assay (IFA) at 48 h post-transfection. pDsRed2-ER contains a coding sequence for red fluorescent protein (RFP) tagged with an N-terminal signal peptide and a C-terminal ER retention sequence (K-D-E-L). pDsRed-Monomer-Golgi encodes a fusion protein of an RFP, while the N-terminal 81 amino acids of human β-1,4-galactosyltransferase contain the membrane-anchoring signal peptide utilized in trans-Golgi network (TGN) localization.

## Proteasome inhibitor treatment

To analyse the effect of MG132, a proteasome inhibitor, on STAT2 degradation, HEK293T cells were transfected with or without NSm-FLAG expression plasmid. At 24 h post-transfection, the cells were treated with or without MG132 (0.7 μM; Sigma-Aldrich) for 16 h prior to harvesting.

## Quantitative real-time RT-PCR

Quantitative real-time RT-PCR was performed as previously described using the primers corresponding to the *OAS1* gene (forward primer, 5′- CATCCGCCTAGTCAAGCACTG-3′; reverse primer, 5′- CACCACCCAAGTTTCCTGTAG-3′), *GAPDH* gene (forward primer, 5′- TGTTGCCATCAATGACCCCTT-3′; reverse primer, 5′- CTCCACGACGTACTCAGCG-3′) or *STAT2* gene (forward primer, 5′- CAGGTCACAGAGTTGCTACAGC-3′; reverse primer, 5′- CGGTGAACTTGCTG CCAGTCTT-3′) [10,12]. Real-time RT-PCR was performed using the One Step TB Green PrimeScript PLUS RT-PCR Kit (TaKaRa). Relative mRNA levels were calculated using the $2^{-\Delta\Delta CT}$ method with GAPDH mRNA used as an internal control and are shown as relative fold-changes normalized to untreated control samples.

## VSIV infection

HEK293T cells transfected with NSm-FLAG or SFTSV NSs-HA were treated or untreated with IFN-αA/D (500 U/mL), which is universal IFN-I, (PBL Assay Science) for 24 h. The treated and untreated cells were infected with VSIV at a multiplicity of infection (MOI) of 10. At 6 h post-inoculation, the cells were lysed in 200 µL of sodium dodecyl sulphate sample buffer.

## Western blotting

Western blotting was performed as previously described using the following antibodies [10,12,13]: anti-HA (clone 3F10; Roche), anti-His (clone 9F2; Fujifilm Wako), anti-FLAG (catalog no. PM020; MBL), anti-VSIV M (clone 23H12; Kerafast), anti-VSIV G (clone P5D4; Sigma-Aldrich), anti-STAT2 (catalog no.: D9J7L; Cell Signaling Technology), anti-STAT1 (catalog no.: D19KY; Cell Signaling Technology), anti-STAT2 (phosphor Y690) (catalog no.: D3P2P; Cell Signaling Technology), anti-STAT1 (phosphor Y701) (catalog no.: D4A7; Cell Signaling Technology), anti-CCHFV Gn (in-house production) or anti-β-actin (catalog no. AC-15; Sigma-Aldrich). The band intensities of VSIV-G, VSIV-M, STAT1, pSTAT1, STAT2, pSTAT2, NSm-FLAG, and β-actin were quantified using ImageJ software (National Institutes of Health). The expression level of VSIV-G, VSIV-M, STAT1, pSTAT1, STAT2, and pSTAT2 ware adjusted by the amounts of β-actin. The STAT1 or STAT2 levels in the IP eluents were adjusted according to the amount of NSm-FLAG.

## Indirect IFA

To observe the co-localization between host and viral proteins, an indirect IFA was performed as previously described [10,12]. All transfected cells were fixed with 4% paraformaldehyde in phosphate-buffered saline (Wako). The fixed cells were incubated in cold methanol for permeabilization and in blocking buffer (5% goat serum (Gibco) and 0.3% Triton X-100 (Wako) in phosphate-buffered saline (Shimadzu Diagnostics)) for blocking. The cells were then treated with primary antibodies overnight at 4°C and stained with secondary antibodies for 2 h at room temperature with 4′,6-diamidino-2-phenylindole (DAPI)(Roche) to visualize the nuclei. Images were acquired using an LSM780 confocal-microscope (Carl Zeiss). Distribution of signals were measured by ImageJ software (National Institutes of Health).

## Coimmunoprecipitation assay

The co-immunoprecipitation (co-IP) assay was performed as previously described [10,12,13]. The cells were mixed with magnetic beads conjugated to anti-FLAG (FLA-1; MBL) or anti-HA (5D8; MBL) monoclonal antibodies and incubated at 4°C for 3 h or overnight. The magnetic beads were then washed with lysis buffer (25 mM Tris-HCl (Wako), 150 mM NaCl (Wako), 1 mM ethylenediaminetetraacetic acid [EDTA] (Wako), and 1% Triton X-100 (Wako)) and wash buffer (50 mM Tris-HCl (Wako), 1% NP-40 (Wako), 0.25% deoxycholic acid sodium salt (Wako), 150 mM NaCl (Wako), and 1 mM EDTA (Wako)) and analysed by western blotting.

## Statistical analyses

The data are expressed as mean and standard deviation (SD), and statistically significant differences were determined using Student's *t* test (Fig 2C), one-way ANOVA with Tukey's (Figs 2B and 6E), or Dunnett'S (Figs 1A–1C and 6F), or two-way ANOVA with Tukey's (Fig 4) multiple comparison test. All statistical analyses were performed using Prism GraphPad 9 software (GraphPad Software, Inc.).

## Results

### CCHFV NSm suppresses IFN-I signaling

IFN-I activates the ISRE promoter and upregulates ISG expression. To investigate whether NSm suppresses IFN-I signaling, we examined the effect of NSm on ISRE activation by IFN-I in HEK293T cells, which are IFN-I-competent and susceptible to CCHFV infection [10,14], using a Dual-Luciferase Reporter Assay kit (Promega). In this experiment, the cells were treated or untreated with IFN-αA/D (500 U/ml) for 18 h. Consistent with previous reports [12], IFN-αA/D induced strong ISRE activation in HEK293T cells (Fig 1A). In this condition, the GP and NSm proteins significantly suppressed the ISRE activation driven by IFN-αA/D, while GP-deleted NSm protein did not. Each protein was detected by western blotting (Fig 1A). In cells expressing NSm, two distinct bands were observed (Fig 1A). The upper band may correspond to a post-translationally modified form of the NSm protein, such as glycosylation or phosphorylation. In addition, NSm dose-dependently inhibited ISRE activation (Fig 1B). To investigate the effect of NSm on ISG mRNA induction by IFN-I, the induction of oligoadenylate synthetase 1 (*OAS1*) mRNA was examined by quantitative real-time RT-PCR. The induction of *OAS1* by IFN-αA/D was inhibited by NSm expression in a dose-dependent manner (Fig 1C).

We used a VSIV infection assay to monitor the antiviral state because its replication is efficiently limited by IFN-I treatment [15]. HEK293T cells transfected with the expression plasmid for NSm-FLAG or SFTSV NSs-HA, which is known to function as an IFN-I signaling inhibitor in HEK293T cells [10,12], were treated with IFN-αA/D. At 24 h post-treatment, the cells were infected with VSIV. At 6 h post-infection, the VSIV G and M protein expressions were confirmed by western blotting. As shown in Fig 1D, the treatment of IFN-αA/D reduced VSIV G and M protein expression by 0% and 14%, respectively. In the presence of NSm or SFTSV NSs, the relative expression level of VSIV G and M protein was 8.4% (NSm) or 21.6% (SFTSV NSs) and 78.9% (NSm) or 77.3% (SFTSV NSs), respectively, indicating that the expression level of VSIV G and M proteins was partially recovered by NSm and SFTSV NSs. These results suggest that CCHFV NSm interferes with IFN-I signaling in human cells.

### CCHFV NSm degrades STAT2 via a ubiquitin/proteasome-dependent pathway

To examine the protein expression levels of STAT1 and STAT2 in the presence of NSm, HEK293T cells were transfected with or without NSm-FLAG expression plasmid. Regardless of the NSm concentration, the STAT1 protein expression levels did not change (Fig 2A). However, the STAT2 protein expression level decreased in an NSm concentration–dependent manner (Fig 2A). In addition, to investigate the STAT1 and STAT2 phosphorylation induced by IFN-I in the presence of NSm, transfected cells were treated or untreated with IFN-αA/D (2000 U/ml) for 30 min. As shown in Fig 2A, NSm downregulated STAT2 but not STAT1 phosphorylation. Moreover, to investigate whether NSm reduce the STAT2 mRNA expression level, HEK293T cells were transfected with or without the expression plasmid of NSm (2.0 µg). The quantitative real-time RT-PCR results showed an increase in STAT2 mRNA levels in the presence of NSm (Fig 2B). These results suggest that NSm downregulates STAT2 at the protein but not mRNA level.

We then examined whether NSm induces STAT2 degradation via a ubiquitin/proteasome-dependent pathway. HEK293T cells were transfected with or without the NSm-FLAG expression plasmid and then treated with proteasome inhibitor MG132 (0.7 µM) for 16 h. NSm reduced the amount of STAT2 by 50% (Fig 2C). In contrast, MG132 restored the amount of STAT2 to the control levels (Fig 2C). K48-linked ubiquitination is a process in which a chain of ubiquitin

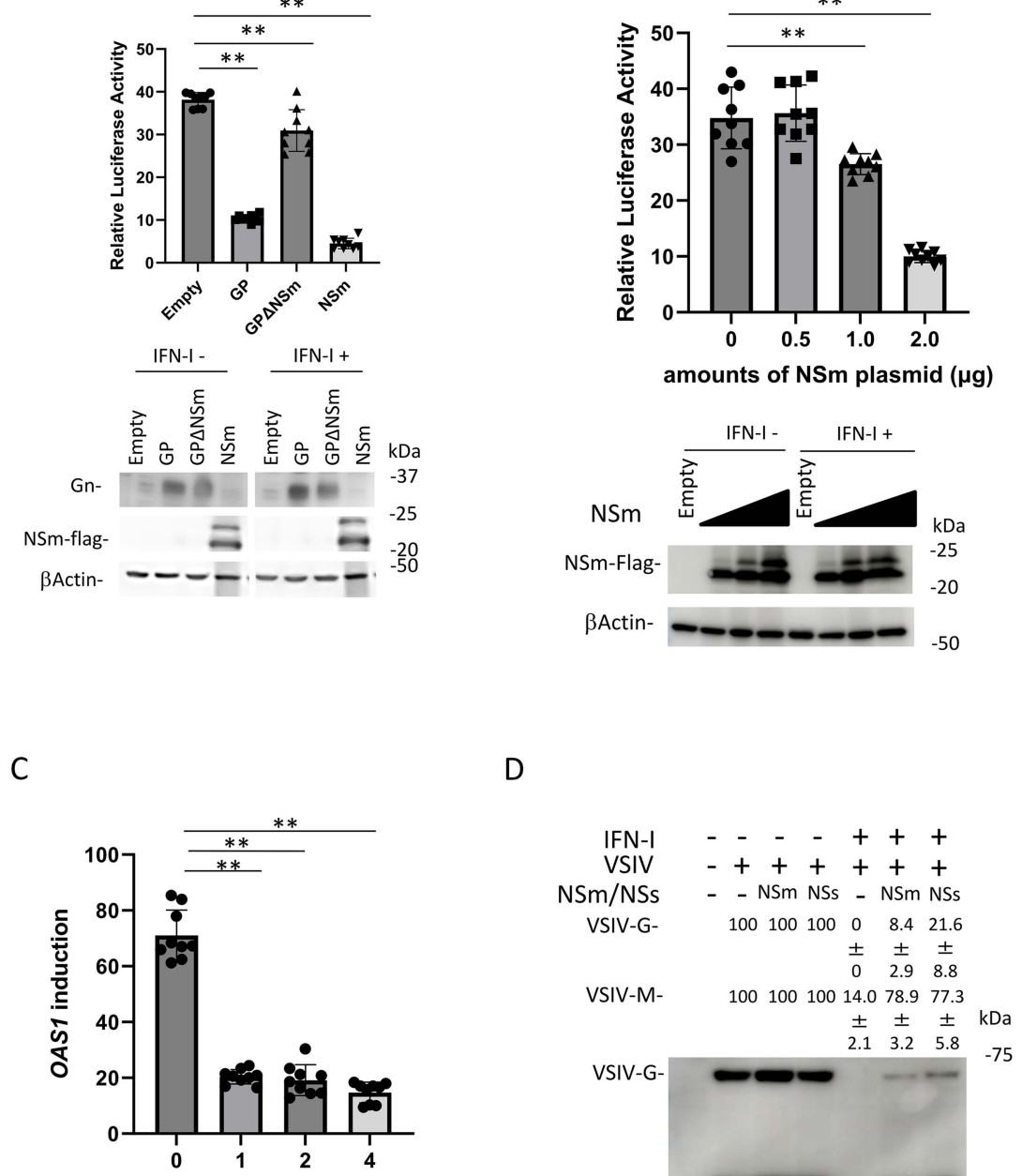

**Fig 1. Function of NSm as an IFN-I antagonist in human cell line.** (A) The reporter plasmids were transfected into HEK293T cells with or without the expression plasmid for CCHFV GP, CCHFV GPΔNSm, or NSm-FLAG (2μg). At 24h post-transfection, the cells were treated with IFN-αA/D (500 U/mL)

or left untreated for 18 h and then lysed for the measurement of luciferase activity and detection of protein expression using immunoblotting. Relative light units in transfected cells with each plasmid in the absence of IFN-αA/D were set as 1. Fold-change activation by IFN-αA/D is indicated. (B) NSm dose-dependent inhibition of ISRE activation was examined using different amounts of the NSm expression plasmid in HEK293T cells. (C) NSm-FLAG expression plasmid was transfected into HEK293T cells. At 24 h post-transfection, the cells were treated with IFN-αA/D (500 U/mL) for 10 h or left untreated. Expression levels of *Oas1* mRNAs in each cell line were measured by quantitative real-time reverse transcription polymerase chain reaction. The mRNA expression levels of *OAS1* in untreated cells were set as 1. Fold-change activation by IFN-αA/D is indicated. (D) The expression plasmids for NSm-FLAG or SFTSV NSs–hemagglutinin were transfected into HEK293T cells. At 24 h post-transfection, the cells were treated with IFN-αA/D (500 U/ mL) for 24 h or left untreated and then infected with VSIV at a multiplicity of infection of 10. At 6 h post-infection, the protein expressions of VSIV M and G were detected by western blotting. The numbers indicate the relative band intensities. The relative expression rate of VSVI G and M in untreated cells with IFN-I was set at 100%. These assays were independently performed in triplicate. The data represent means and standard deviations. *$P < 0.05$; **$P < 0.01$ versus empty. Blots have been cropped; full uncropped blots are available as S1Fig.CCHFV, Crimean-Congo hemorrhagic fever virus; GP, glycoprotein precursor; IFN-I, type I interferon; VSIV, vesicular stomatitis Indiana virus.

molecules is specifically linked via the lysine 48 (K48) residue of ubiquitin and added to a target protein [16]. This modification primarily functions as a signal for protein degradation by the proteasome. Therefore, we performed a co-IP assay using the lysates of HEK293T cells transfected with K48-linked ubiquitin, STAT2-FLAG, or NSm-His expression plasmids and treated with proteasome inhibitor MG132 (0.7 μM) for 16 h. The co-IP assay using anti-HA antibodies indicated that the amount of K48 ubiquitin-modified STAT2 was increased by NSm. These results indicate that NSm induced STAT2 degradation via a ubiquitin/proteasome-dependent pathway.

## CCHFV NSm interacts with STAT2

To examine the interaction of NSm with STAT1 and STAT2, we performed co-IP assays using the lysates of HEK293T cells transfected with the NSm-FLAG expression plasmid. As shown in Fig 3A, co-IP of STAT2 with NSm was observed, whereas co-IP of STAT1 with NSm was not. In addition, strong binding of NSm to STAT2 was observed after MG132 treatment (Fig 3A). To understand the difference in the binding ability of NSm to STAT1 and STAT2, we transfected the expression plasmid for NSm-FLAG with the expression plasmids for STAT1-His or STAT2-His into HEK293T cells and then performed a co-IP assay. In an experiment demonstrating the interaction of NSm with endogenous STAT2 (Fig 3A), NSm efficiently bound to STAT2, whereas its interaction with exogenous STAT1 was very weak (Fig 3B). The interactions of NSm with STAT1 and STAT2 were further investigated by the subcellular co-localization of NSm, STAT1, and STAT2. The NSm-FLAG expression plasmid was co-transfected with the STAT1- or STAT2-GFP expression plasmid into HEK293T cells, and the subcellular localization of NSm, STAT1, and STAT2 was observed by indirect IFA. NSm was co-localized with STAT1 and STAT2 (Fig 3C). To further confirm that STAT2 was the target protein of NSm, the anti-IFN-I signaling activity of NSm was evaluated in STAT2-overexpressing HEK293T cells (Fig 4). As done in Fig 1A and 1B, the transfected cells were treated or untreated with IFN-αA/D (500 U/ml) for 18 h. The expression of each protein was observed by immunoblotting. In this condition, the inhibition of ISRE activity by NSm was restored by STAT2 overexpression. Therefore, we suggest that the interaction between NSm and STAT2 inhibits IFN-I signaling.

## NSm alters subcellular STAT2 localization

NSm was previously reported to localize to the endoplasmic reticulum (ER) and Golgi apparatus [14]. Therefore, here we hypothesized that the subcellular localization of STAT2 is altered by its binding to NSm. To address this possibility, STAT1-His or STAT2-His were co-expressed with fluorescent marker proteins that localize in the ER or TGN. Consistent with a previous report [17], NSm co-localized with both ER and TGN RFP marker proteins (Fig 5B, 5D, 5F, and 5H). In the absence of NSm, STAT2 was localized in the cytoplasm and ER but not in the TGN (Fig 5E and 5G). In contrast, STAT2 was partially localized to the TGN in the presence of NSm (Fig 5H). Although we did not perform quantitative co-localization analysis, representative images showing the spatial distribution of each signal were provided. Based on these

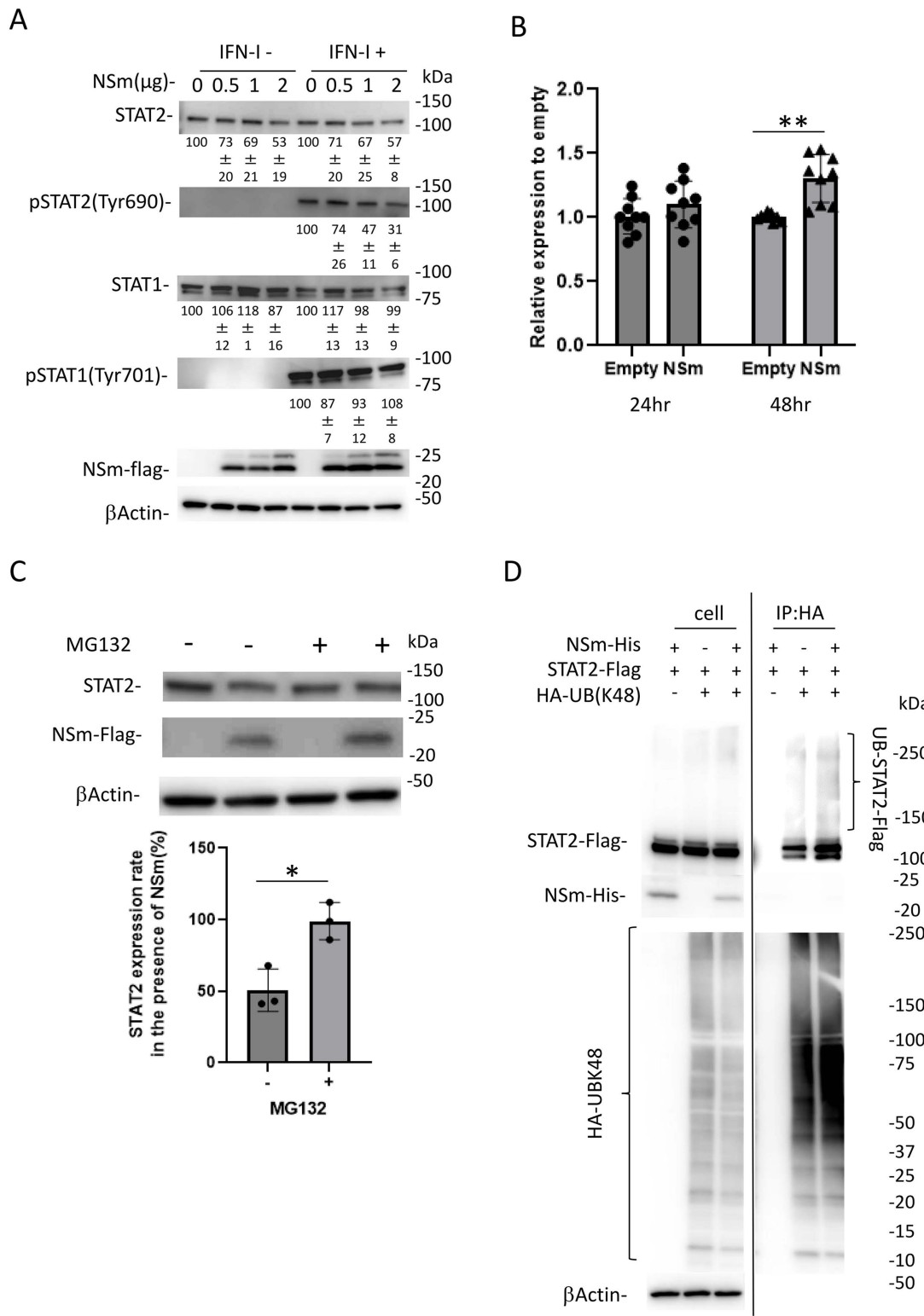

**Fig 2. NSm degrades STAT2 via a ubiquitin/proteasome-dependent pathway.** (A) HEK293T cells transfected with the expression plasmid for NSm-FLAG and treated with IFN-αA/D (2,000 U/mL) for 30 min or left untreated and then lysed for the detection of expression of each protein by immunoblotting. The numbers indicate the relative band intensities. The relative expression rate of each protein in mock-transfected cells was set at 100%. (B) The

expression plasmid for NSm-FLAG (2 µg) was transfected into HEK293T cells. At 24 or 48 h post-transfection, the RNA expression level of STAT2 was determined by quantitative real-time reverse transcription polymerase chain reaction. (C)The transfected HEK293T cells with or without the expression plasmid for NSm-FLAG were treated with MG132 (0.7 µM) for 16 h or left untreated. The expression of each protein was detected by western blotting (upper panel). The expression level of STAT2 was quantified (lower panel). The relative expression rate of STAT2 in mock-transfected cells was set at 100%. (D) HEK293T cells were transfected with STAT2-FLAG, NSm-His, and ubiquitin (K48)-hemagglutinin, and co-immunoprecipitation assays were performed by anti-hemagglutinin antibodies. These assays were independently performed in triplicate. The data are shown as means and standard deviations. *$P < 0.05$; **$P < 0.01$ versus empty (B) or no treatment (C) conditions. Blots have been cropped; full uncropped blots are available as S2 and S3 Figs.

observations, we described STAT2 as being partially localized to the TGN under NSm-expressing conditions. STAT1 was localized in the nucleus, cytoplasm, and ER regardless of NSm expression (Fig 5A–5D). These findings indicate that NSm alters the subcellular localization of STAT2 but not STAT1.

## STAT2 SH domain is important for NSm binding

To investigate the region of STAT2 binding to NSm, we prepared a series of STAT2 deletion mutants (Fig 6A) and examined their binding using a co-IP assay. As shown in Fig 6B, the expression level of Δ5 was lower than that of the others. The cause of this is thought to be the instability of protein due to deletion. In this condition, NSm bound to STAT2Δ1, Δ2, Δ3, and Δ4, but not Δ5 (Fig 6B). However, it was still unclear whether the binding of NSm and STAT2 was mediated by SH domain, since the expression level of Δ5 was extremely low. Therefore, we also prepared two chimeric proteins between STAT1 and STAT2: STAT1(2SH) and STAT2(1SH) (Fig 6C) and performed a co-IP assay to examine the interaction between NSm and both chimeric proteins (Fig 6D). Although the expression level of the STAT1(2SH) mutant was much lower than that of STAT1 and STAT2, co-immunoprecipitation of STAT1(2SH) with NSm was observed (Fig 6D). Contrary to our expectations, we found that STAT2(1SH) also interacted with NSm (Fig 6D). Therefore, to quantitatively evaluate these results, we normalized the expression levels of STAT1, STAT2, and both chimeric proteins in the co-IP solution to those in the whole-cell lysates. As shown in Fig 6E, the binding ability of STAT1(2SH) to NSm was much stronger than that of STAT1 to NSm, while that of STAT2(1SH) to NSm was weaker than that of STAT2 to NSm.

Next, we quantitatively evaluated changes in protein expression levels in whole cells using NSm expression. In the presence of NSm, the STAT2 and STAT1(SH) expression levels decreased, whereas the STAT1 and STAT2(1SH) expression levels did not (Fig 6F). These results suggest that the STAT2 SH domain is important for interaction with and degradation by NSm.

## Discussion

This study found that the interaction of NSm with STAT2 inhibited IFN-I signaling in a human cell line. Although the STAT family of proteins is highly conserved among mammals including humans and mice, only STAT2 is highly divergent [18]. Intriguingly, STAT2 is involved in determining the species specificity of several viruses. For example, we recently reported that the SFTSV NSs protein targets human but not murine and hamster STAT2 [12]. In addition, SFTSV infections do not cause severe disease in immunocompetent mice and Syrian hamsters, whereas SFTSV infection is lethal to IFN-I receptor– or STAT2-knockout mice and Syrian hamsters [12,19]. In addition, CCHFV causes severe disease in humans and STAT2-deficient mice and Syrian hamsters but not in wild-type mice and Syrian hamsters [20,21]. Therefore, CCHFV may not be able to replicate in mice and Syrian hamsters due to NSm being unable to antagonize murine and hamster STAT2.

Innate immunity induced by viral infection can be divided into IFN-I induction and signaling phases. CCHFV L contained an Otubain-like cysteine protease (OTU) domain. CCHFV OTU suppresses IFN-I induction by cleaving ubiquitin protein of retinoic acid–inducible gene I [22]. Therefore, CCHFV can inhibit IFN-I induction and signaling. Like CCHFV, SFTSV [10,23], Ebola virus [24,25], dengue virus [26], and severe acute respiratory syndrome coronavirus 2 [27], which are highly pathogenic to humans, suppress IFN-I induction and signaling in humans. In contrast, the Uukuniemi virus, a

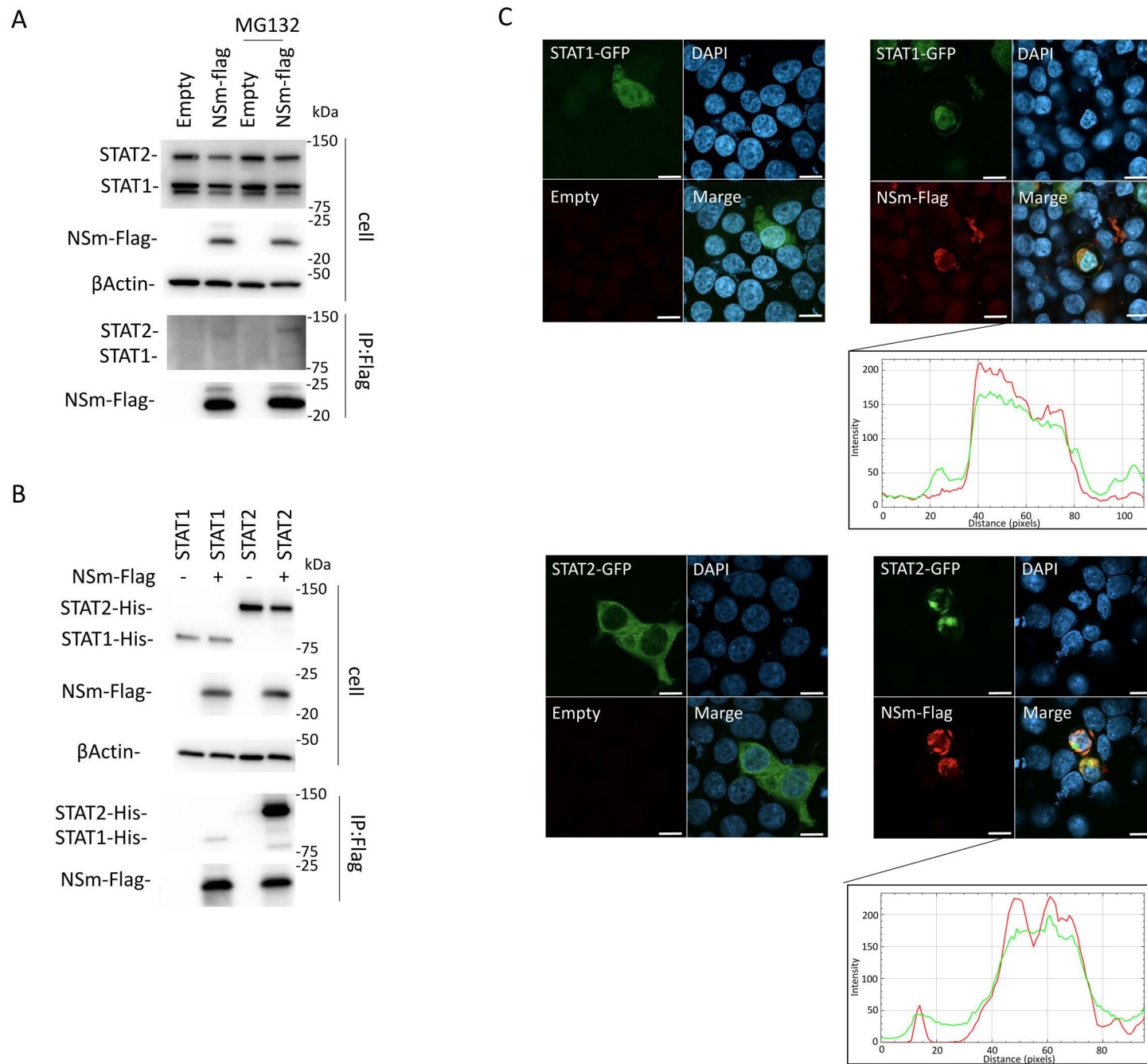

**Fig 3. Interaction of NSm with STAT1 and STAT2.** (A) HEK293T cells were transfected with the expression plasmid for NSm-FLAG, and co-immunoprecipitation assays were performed. (B) HEK293T cells were transfected with the expression plasmid for NSm-FLAG and STAT1-His or STAT2-His. The protein expression levels in cell lysates (upper) and co-immunoprecipitation assays (lower) using an anti-FLAG antibody are shown. Blots have been cropped; full uncropped blots are available as S4 Fig. (C) Co-localization of NSm with STAT1 and STAT2. HEK293T cells were transfected with the expression plasmid for NSm-FLAG (0.1 μg) and STAT1- or STAT2-GFP (0.1 μg). An immunofluorescent assay was also performed with NSm, STAT1 (or STAT2), and the nuclei shown in red, green, and blue, respectively. The distribution of NSm and STAT1 (or STAT2) was shown. Representative results of western blotting (A and B) and immunofluorescence (C) assays are shown. The scale bar represents 20 μm.

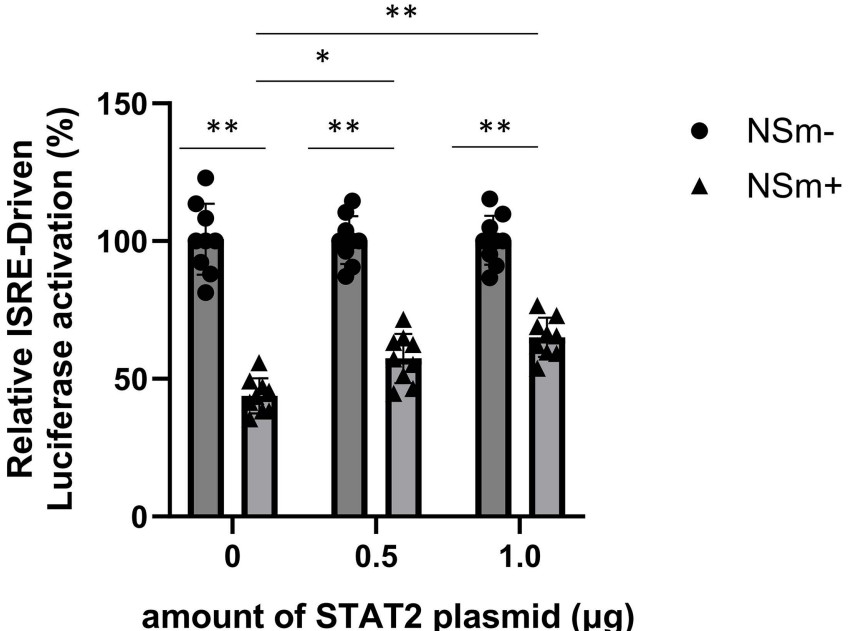

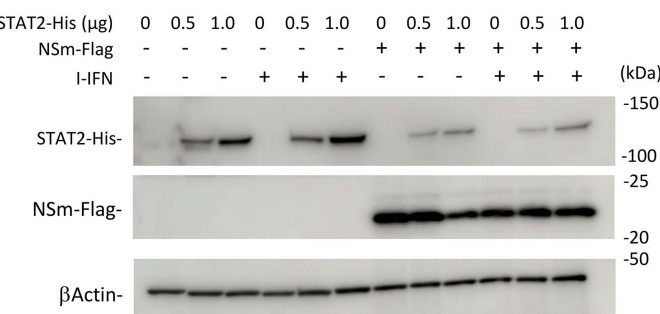

**Fig 4. Function of NSm in STAT2-overexpressing HEK293T cells.** The reporter plasmids were transfected into HEK293T cells with the expression plasmid for NSm-FLAG and STAT2-His. At 24 h post-transfection, the cells were treated with IFN-αA/D (500 U/mL) for 18 h or left untreated and then lysed for the measurement of luciferase activity (upper panel). The interferon (IFN)-stimulated response element (ISRE) activity was calculated by dividing relative light units in IFN-αA/D-treated cells by those of cells not treated with IFN-αA/D. The ISRE activity in the absence of NSm was set as 100%. The expression of each protein was shown by immunoblotting (lower panel). Blots have been cropped; full uncropped blots are available as S5 Fig. The assays were independently performed in triplicate. The data represent means and standard deviations. * $P<0.05$; **$P<0.01$ versus no NSm or STAT2. Blots have been cropped; full uncropped blots are available as S5 Fig.

non-pathogenic virus in humans, suppresses IFN-I induction but not IFN-I signaling [23]. These findings suggest that both anti-IFN-I induction and anti-IFN-I signaling are required for the virus to be highly pathogenic to humans.

In this study, we demonstrated that NSm bound to STAT2 (Fig 3A and 3B). However, the binding ability of NSm to endogenous STAT2 was less than that to exogenous STAT2. This discrepancy may be explained by the differences in the protein expression level between exogenous STAT2 and endogenous STAT2. Although experiments using expression plasmids do not necessarily mimic the endogenous protein expression levels in the cell, both the expression level of endogenous and exogenous STAT2 was reduced by NSm (Fig 3A and 3B). These results indicate that NSm specifically interacts with STAT2. Animal experiments using mice, such as human STAT2 knock-in mice, would be required to

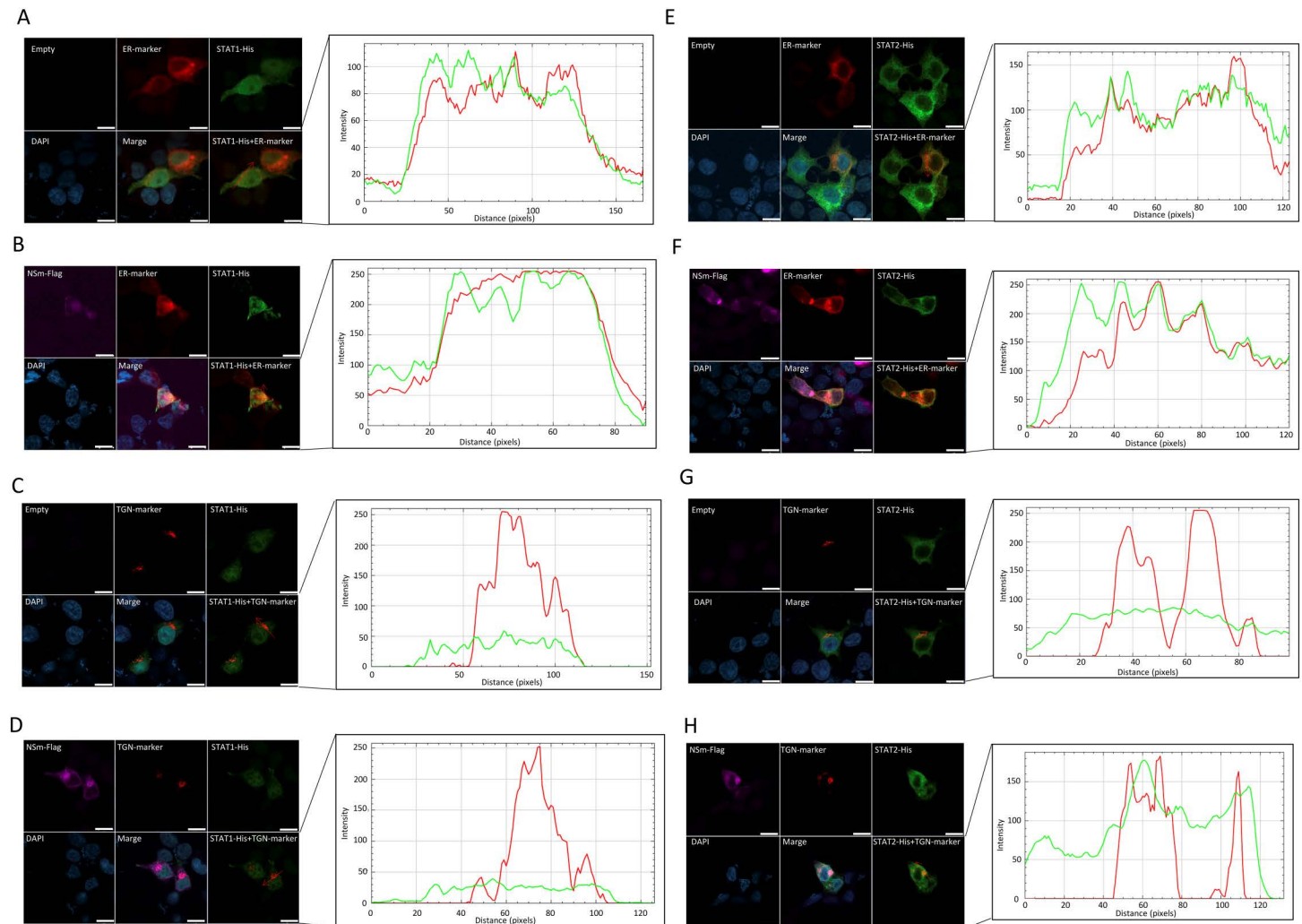

**Fig 5. Effect of NSm on STAT1 and STAT2 localization.** HEK293T cells were transfected with the expression plasmid for NSm-FLAG (0.1 μg), STAT1- or STAT2-His (0.1 μg), and each marker (0.1 μg). Co-localization of STAT1 and endoplasmic reticulum (ER) marker (A) or trans-Golgi network (TGN) marker (B) in the absence of NSm. Co-localization of STAT2 and ER marker (C) or TGN marker (D) in the absence of NSm. Co-localization of STAT1 and ER marker (E) or TGN marker (F) in the presence of NSm. Co-localization of STAT2 and ER marker (G) or TGN marker (H) in the presence of NSm. HEK293T cells were transfected with the expression plasmid for NSm-FLAG and STAT1-His or STAT2-His and pDsRed2-ER (ER marker) or pDsRed-Monomer-Golgi (TGN marker). An immunofluorescence assay was also performed with NSm, STAT1 (or STAT2), and ER marker (or TGN marker), and the nuclei are shown in purple, green, red, and blue, respectively. The distribution of STAT1 or STAT1 and ER marker or TGN marker was shown. Representative results of the immunofluorescence assay are shown. The scale bar represents 20 μm.

investigate whether STAT2 degradation activity by NSm is involved in CCHFV pathogenicity and fully understand the relationship between STAT2 and NSm *in vivo*.

Interestingly, we demonstrated that NSm increased mRNA expression level of STAT2 (Fig 2B). Based on our observations, we hypothesize that NSm may play a role in modulating host RNA expression. Although direct evidence is currently lacking, the changes in cellular localization and signaling pathways observed in NSm-expressing cells suggest a potential regulatory function. Further studies will be required to elucidate the precise mechanisms involved.

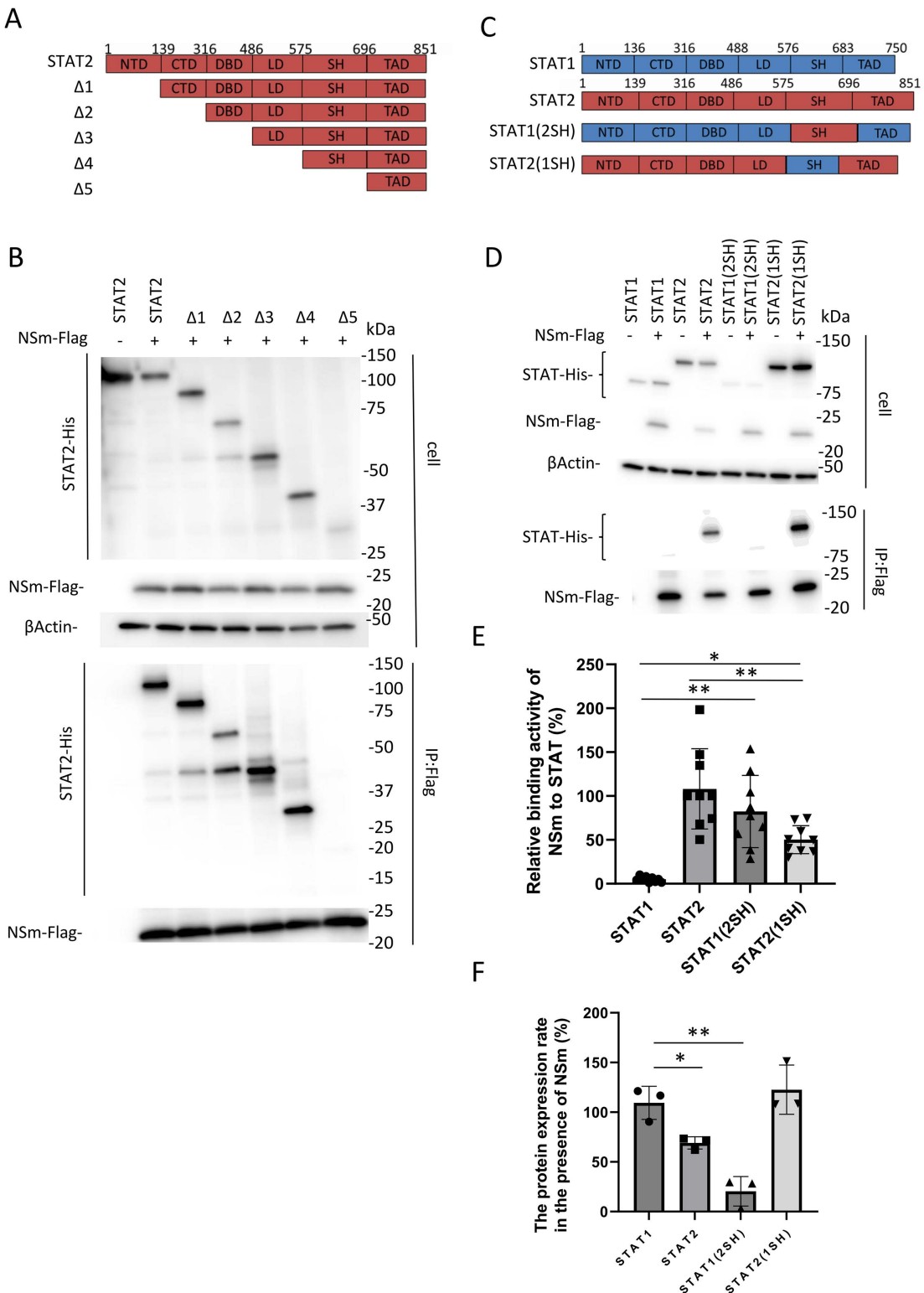

**Fig 6. Determination of the region of STAT2 for binding to NSm.** (A) Schematic representation of the deletion mutants of STAT2. (B) HEK293T cells were co-transfected with the expression plasmid for NSm-FLAG and each of the deletion mutants, following which co-immunoprecipitation assays were performed. (C) Schematic representation of the chimeric STAT1 and STAT2 mutants. (D) HEK293T cells were co-transfected with the expression

plasmid for NSm-FLAG and each of the chimeras. The protein expression levels in cell lysates (upper) and in co-immunoprecipitation assays (lower) using an anti-FLAG antibody are shown. (E) The binding activity of NSm to each STAT was quantified. (F) The protein expression level of each STAT in the presence of NSm was calculated from the band intensity in the cell lysates (upper panel in D). These assays were independently performed in triplicate. Representative results of western blotting assays (B and D) are shown. The data represent means and standard deviations (E and F). *$P < 0.05$; **$P < 0.01$ versus STAT1(2SH) or STAT2(1SH) (E). *$P < 0.05$; **$P < 0.01$ versus STAT1 (F). Blots have been cropped; full uncropped blots are available as S6 and S7 Figs.

This study's findings indicated that STAT2 degradation by NSm occurs via a ubiquitin-mediated pathway. The search for motifs related to protein degradation in the NSm sequence by Eukaryotic Linear Motif (ELM) resource (http://elm.eu.org/) resulted in the identification of the DEG_Nend_UBRbox_1 and the SPOP motif. The DEG_Nend_UBRbox_1 motif is a known N-terminal motif that initiate protein degradation by binding to the UBR-box of N-recognins and is related to the N-end rule pathway, which regulates protein stability by targeting proteins for ubiquitin-dependent proteasomal degradation [28]. SPOP is an adaptor protein of the CUL3-RING E3 ubiquitin ligase complex that recognizes specific substrates and promotes their ubiquitination and proteasomal degradation [28]. Therefore, NSm may degrade STAT2 by recruiting these host factors. In addition, these motifs are conserved among the various CCHFV strains (Fig 7).

Taken together, our findings indicate that the interaction of NSm and STAT2 inhibits IFN-I signaling. Elucidation of the evasion mechanisms of the innate immunity of viruses will contribute to our understanding of virus attenuation and facilitate the development of live attenuated vaccines. Moreover, molecular compounds that antagonize the interaction between viral proteins such as NSm and host factors such as STAT2 may be used to treat viral infections.

DEG_Nend_UBRbox_1

```
CCHFV Hoti (Europe 1) [used in this study]   1 RKLLQVSESTGVALKRSSWLIVLLVLLTVSLSPVQSAPVGHGKTIETYQTREGFTSICLF 60
CCHFV ArD8194 senegal 1969 (Africa 1)        1 ................C.M.T..I.....M..........KRAV.V..M..SY.G.... 60
CCHFV Beruwe 2008 (Africa 2)                  1 .R.........M......C.MTT..I..V...........QE.AV.V..V..SY..M... 60
CCHFV IbAr10200 (Africa 3)                    1 .......................F...........I.Q.....A.RA...Y...... 60
CCHFV NIV161064 (Asia 1)                      1 ...........TV.........F...........I........RV..EY...... 60
CCHFV NIVA118594 (Asia 2)                     1 ...........I........F...IT.....I......A.RA..EY...... 60

CCHFV Hoti (Europe 1) [used in this study]  61 MLGSILFIVSCLVKGLVDSVSDSFFPGLSVCKTCSIGSINGFEIESHKCYCSLFCCPYCR 120
CCHFV ArD8194 senegal 1969 (Africa 1)       61 V...V..A..W...A.I..IGN.......I........................... 120
CCHFV Beruwe 2008 (Africa 2)                 61 V...V..A..W.I...I.GIGN........................... 120
CCHFV IbAr10200 (Africa 3)                   61 V...........M........GN......I.....S.................. 120
CCHFV NIV161064 (Asia 1)                     61 V.........F.M.....G.GNI....................... 120
CCHFV NIVA118594 (Asia 2)                    61 V...V..V....M.......GNI......FI.................. 120

CCHFV Hoti (Europe 1) [used in this study] 121 HCSADREIHQLHLNICKKRKTGSNVMLAVCKRMCFRATIEASRRALLIRSIINTTFVICI 180
CCHFV ArD8194 senegal 1969 (Africa 1)      121 A..S.KIT.RM...V.....V................K......N..TF..N...S...... 180
CCHFV Beruwe 2008 (Africa 2)               121 A..S.KIT.RM...V.....A............K..V...NT....G...S.....V 180
CCHFV IbAr10200 (Africa 3)                 121 ...T.K...K...S......K.........L......M.V.N...F...........L.. 180
CCHFV NIV161064 (Asia 1)                   121 .....G.......S................M.V.NK..F.........V.. 180
CCHFV NIVA118594 (Asia 2)                  121 ............S..................T.M.V.NK..FV.........V.. 180

CCHFV Hoti (Europe 1) [used in this study] 181 LTLTICVVSTSA                                                  192
CCHFV ArD8194 senegal 1969 (Africa 1)      181 .I.V..........                                                192
CCHFV Beruwe 2008 (Africa 2)               181 .I.A..........                                                192
CCHFV IbAr10200 (Africa 3)                 181 .I.AV.......                                                  192
CCHFV NIV161064 (Asia 1)                   181 .I.AV.......                                                  192
CCHFV NIVA118594 (Asia 2)                  181 .I.AV.......                                                  192
```

SPOP motif

**Fig 7. Sequence comparison of the NSm of CCHFV.** Sequence alignment of the NSm was conducted based on the amino acid sequences of representative CCHFV strain. DEG_Nend_UBRbox_1 and SPOP motif are indicated by a box. The genotype of each strain is shown in parentheses.

## Supporting information

**S1 Fig. (A) Original (uncropped) blots of** Fig 1A. (B) Original (uncropped) blots of Fig 1B. (C) Original (uncropped) blots of Fig 1C. (D) Original (uncropped) blots of Fig 1D. Cropped regions are indicated by red squares.
(TIF)

**S2 Fig. (A) Original (uncropped) blots of** Fig 2A. (B) Original (uncropped) blots of Fig 2C. Cropped regions are indicated by red squares.
(TIF)

**S3 Fig. Original (uncropped) blots of** Fig 2D. Cropped regions are indicated by red squares.
(TIF)

**S4 Fig. (A) Original (uncropped) blots of** Fig 3A. (B) Original (uncropped) blots of Fig 3B. Cropped regions are indicated by red squares.
(TIF)

**S5 Fig. Original (uncropped) blots of** Fig 4 **(lower panel).** Cropped regions are indicated by red squares.
(TIF)

**S6 Fig. Original (uncropped) blots of** Fig 6B. Cropped regions are indicated by red squares.
(TIF)

**S7 Fig. Original (uncropped) blots of** Fig 6D. Cropped regions are indicated by red squares.
(TIF)

**S1 Data. Excel sheet containing raw data of figures.**
(XLSX)

## Acknowledgments

We are grateful to all members of the Department of Emerging Infectious Diseases, Institute of Tropical Medicine, Nagasaki University.

## Author contributions

**Conceptualization:** Rokusuke Yoshikawa, Jiro Yasuda.

**Data curation:** Rokusuke Yoshikawa.

**Formal analysis:** Rokusuke Yoshikawa.

**Funding acquisition:** Rokusuke Yoshikawa, Jiro Yasuda.

**Investigation:** Rokusuke Yoshikawa, Yasuteru Sakurai, Sayako Kondo, Mayuko Kimura.

**Methodology:** Rokusuke Yoshikawa.

**Project administration:** Jiro Yasuda.

**Resources:** Rokusuke Yoshikawa.

**Supervision:** Jiro Yasuda.

**Validation:** Jiro Yasuda.

**Visualization:** Rokusuke Yoshikawa.

**Writing – original draft:** Rokusuke Yoshikawa.

**Writing – review & editing:** Rokusuke Yoshikawa, Jiro Yasuda.

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
