## [Decision Letter · Decision Letter 0]

2 Jul 2025

Crimean-Congo hemorrhagic fever virus NSm protein inhibits the type I interferon signaling by binding to STAT2.

Dear Dr. Yasuda,

Thank you for submitting your manuscript to PLOS Neglected Tropical Diseases. After careful consideration, we feel that it has merit but does not fully meet PLOS Neglected Tropical Diseases's publication criteria as it currently stands. Therefore, we invite you to submit a revised version of the manuscript that addresses the points raised during the review process.

Please submit your revised manuscript within 60 days Aug 31 2025 11:59PM. If you will need more time than this to complete your revisions, please reply to this message or contact the journal office at plosntds@plos.org. Please include the following items when submitting your revised manuscript:

We look forward to receiving your revised manuscript.

Kind regards,

Roger Hewson

Guest Editor

David Safronetz

Section Editor

Shaden Kamhawi

co-Editor-in-Chief

Paul Brindley

co-Editor-in-Chief

**Additional Editor Comments :**

Subject: Manuscript PNTD-D-25-00630 – Decision: Major Revision Required

Dear Prof Yasuda,

Please accept our apologies for the delay in returning a decision on your manuscript, “Crimean-Congo hemorrhagic fever virus NSm protein inhibits the type I interferon signaling by binding to STAT2” (PNTD-D-25-00630). It has proved challenging to secure timely expert reviews, given the specialised nature of the work.

We have now received two independent peer reviews, both of which recognise the novelty and potential significance of your findings - particularly in attributing a function to the CCHFV NSm protein and its role in disrupting type I interferon signalling via STAT2 degradation. The manuscript is generally well written and addresses an important gap in the understanding of CCHFV immune evasion mechanisms.

However, both reviewers identified a number of substantive issues that will need to be addressed before the manuscript can be considered for publication. These include:

• A need for clearer validation of key findings, particularly those relying on overexpression systems. Several figures (e.g., 1D, 2A, 3A) require repetition or additional quantification.

• Inconsistencies in experimental design and data presentation, including variable NSm expression, unclear interferon treatment protocols, and low-resolution immunofluorescence images.

• Lack of methodological detail in figure legends, limited use of quantitative image analysis, and missing full Western blot data to support conclusions.

• The co-immunoprecipitation results supporting NSm-STAT2 binding are considered modest and may benefit from confirmation under more physiological conditions.

• Further justification for the use of HEK293T cells as a model system and consideration of additional cell lines to improve biological relevance.

In addition to the peer reviews, I have also reviewed the manuscript and would encourage you to consider the following points during revision:

• Explore opportunities to validate STAT2 degradation or NSm-STAT2 interactions in the context of authentic CCHFV infection (e.g., using WT vs ΔNSm virus), or in a reverse genetics system if possible.

• Consider testing species-specific STAT2 interactions to support the hypothesis that NSm contributes to host range restriction.

• Strengthen the evidence for STAT2 re-localisation using quantitative co-localisation analysis and higher-resolution microscopy.

• If proposing a role for NSm in E3 ligase recruitment, include supporting bioinformatic analysis (e.g., motif scans), or clearly frame this as a hypothesis.

• A figure showing NSm sequence conservation across diverse CCHFV strains would help contextualise its functional relevance.

A full list of reviewer comments is enclosed and should be addressed point-by-point in your revision.

Given the nature of the revisions required, we are classifying this decision as Major Revision. We hope that with the necessary improvements, your manuscript can proceed toward publication in PLOS Neglected Tropical Diseases.

Please do not hesitate to get in touch if anything in the reviewer reports is unclear, or if you would like to discuss any specific revisions further.

Best regards,

Guest Editor

**Journal Requirements:**

**Reviewers' Comments:**

Reviewer's Responses to Questions

**Key Review Criteria Required for Acceptance?**

**Methods**

-Are the objectives of the study clearly articulated with a clear testable hypothesis stated?

-Is the study design appropriate to address the stated objectives?

-Is the population clearly described and appropriate for the hypothesis being tested?

-Is the sample size sufficient to ensure adequate power to address the hypothesis being tested?

-Were correct statistical analysis used to support conclusions?

-Are there concerns about ethical or regulatory requirements being met?

Reviewer #1: The aim and approach of the study are clearly explained and presented.

Reviewer #2: Clarity of IFN treatment will help.

State what is IFN A/D.

**Results**

-Does the analysis presented match the analysis plan?

-Are the results clearly and completely presented?

-Are the figures (Tables, Images) of sufficient quality for clarity?

Reviewer #1: There are multiple issues regarding the presented data.

Figure 1D: Although the effect of SFTSV NSs (used as a positive control) is clear, the inhibitory activity of NSm is not convincing. It could be due to low level of NSm expression. I encourage the authors to repeat the experiment and show quantitative data.

Figure 2A and lines 207-8 “the STAT2 protein expression level decreased in an NSm concentration–dependent manner”: It is not convincing with the presented image. The band intensity of STAT2 is very weak compared to STAT1 unlike the image shown in Fig 3A. The data should be repeated and quantitated. The same for phosphorylated STAT2.

Figure 2B and lines 211-4” The quantitative real time RT-PCR results indicated that STAT2 mRNA levels were comparable in the presence and absence of NSm in HEK293T cells (Fig. 2B). These results suggest that NSm downregulates STAT2 at the protein but not mRNA level”: It is unclear how much cDNAs were transfected in this experiment. The authors need to use the same dose of cDNAs to compare their impact on protein and mRNA expression.

Figure 2D: There are a couple of problems in this figure. 1) In the cell lysate (left), there is no difference in STAT2-Flag expression between NSm-His expressing and not expressing cells. 2) In IP material, why more STAT2-Flag (UB unconjugated) was detected in NSm-His expressing cells than non-expressing cells? Was this experiment conducted in the presence of MG132? The authors should consider performing IP using anti-Flag Ab to collect STAT2-Flag and detect UB(K48) by anti-HA Ab.

Figure 3C and line 240 ”NSm was co-localized with STAT1 and STAT2”. : The IF image using conventional microscope at this magnification does not provide convincing data for protein co-localization. Analysis using confocal microscope is recommended.

Figure 4: Western blot data showing the overexpression of STAT2 should be included in this figure.

Figure 5: As explained above, the presented images at this resolution are not convincing to make conclusion stated in lines 250-256.

Figure 6B: Are these experiments conducted in the presence of MG132? The STAT2 level in the lysate seems to be the same regardless the presence of NSm-Flag.

Figure 6F: the data seem to disagree with the image shown in Figure 6D.

Reviewer #2: All figure legends will benefits from more details including controls and quantification methodology.

Results need to be presented with more nuances and address limitations observed.

Supplementary figures should include full blot.

**Conclusions**

-Are the conclusions supported by the data presented?

-Are the limitations of analysis clearly described?

-Do the authors discuss how these data can be helpful to advance our understanding of the topic under study?

-Is public health relevance addressed?

Reviewer #1: The conclusion is not fully supported by the presented data as explained above.

Reviewer #2: Conclusions are not including any limitations or discrepancies observed in the results part.

**Editorial and Data Presentation Modifications?**

Reviewer #1: Figure 2A: The doses of transfected NSm cDNA used for the experiment should be stated in the figure or figure legend.

Reviewer #2: (No Response)

**Summary and General Comments**

Reviewer #1: The finding that CCHFV NSm suppresses IFN-I signaling by targeting STAT2 through ubiquitination is novel and highly significant in the field. The manuscript is written well in general, but it can be improved by providing convincing data as explained above.

Reviewer #2: Yoshikawa and co-authors investigate the role of the CCHFV NSm protein in antagonizing type I interferon (IFN-I) signaling in human cells (293T cells). Using a combination of luciferase reporter assays, co-immunoprecipitation, Western blotting, subcellular localization, and mutagenesis approaches, they demonstrate that NSm binds to human STAT2, promotes its proteasomal degradation, and suppresses IFN-I signaling. These findings suggest that NSm contributes to CCHFV immune evasion and species tropism. While interesting, the manuscript is not addressing many limitations of the study system or results observed, often over-concluding.

Major comment:

The manuscript concludes that NSm interacts with STAT2; however, the co-immunoprecipitation (co-IP) data presented in Figure 3A show only a modest level of STAT2 pull-down, with no clear enrichment compared to the input lysate. While Figure 3B demonstrates a stronger interaction, this result is based on overexpression of STAT2, which may not reflect physiological conditions. The authors should address this discrepancy and discuss the limitations of relying on overexpression systems, particularly in interpreting the strength and specificity of the NSm–STAT2 interaction.

The author used various protocol for IFN treatment, those variations are not stated in the text and make the figure/result part difficult to interpret for the reader. I.e Figure 1A is a WB performed at 18h post-IFN treatment, when Figure 2A is 30 min after IFN treatment.

Could the author explained the rationale behind the use of HEK293T cells (which lack endogenous IFN production and are not the natural targets of CCHFV.), why they did not choose IFN competent cells like A549? or type I IFN deficient cells like Vero E6 to make a comparison?

Could the author explained the low level of NSm expression in HEK293T cells? Could we have an additional figures showing level of expression on >100 cells?

Discrepancies between many blot concerning NSm-FLAG expression, i.e Fig 1A and 1B IFN- and IFN + show very different band intensity in those 2 blots? No mention of this in the results part or in the discussion.

Supplementary data of full blots should be provided for all Western Blot presented here as some discrepancies are observed (including in the same size band e.g; Figure2A Nsm-FLAG empty is white when background of other band is grey)

Minor comments:

Figure 1D: Could author explained the low level of NSm-FLAG? (When compare to Figure 1A this looks strange).

Figure 2B: could the increase of mRNA level at 48h be explained in the text?

Figure 2D: make the labelling of the panel clearer is the lower part HA mediated Immunoprecipitation?

Figure 6B: Could the author address the low level of expression of STAT2 DELTA5? And consequence for the pull down?

Line 116: what is LT-1 cells?

Line 145: What is IFN A/D?

Line 194: could you add a reference regarding the type I IFN treatment

Line 202: Could you address the low level of NSm-FLAG expression in cells?

Line 220: Could you add a reference fro the protein degradation statement?

Line 222-224: Could the author described the results observed, was hard as non-specialist to read that panel without guidance

PLOS authors have the option to publish the peer review history of their article (what does this mean? ). If published, this will include your full peer review and any attached files.

**Do you want your identity to be public for this peer review?** For information about this choice, including consent withdrawal, please see our Privacy Policy .

Reviewer #1: No

Reviewer #2: **Yes: ** Marine J. Petit

**Figure resubmission:**

**Reproducibility:**



---

## [Decision Letter · Decision Letter 1]

21 Oct 2025

Dear Dr. Yasuda,

Thank you for submitting your manuscript to PLOS Neglected Tropical Diseases. Following the receipt of the updated comments from Reviewers 1 and 2 (see below) including my own close reading of the revised manuscript, I am satisfied that all major issues raised during the initial round of peer review have been appropriately addressed. The revised version provides new experimental data where requested, clarifies the mechanism of STAT2 degradation and includes useful additions to the discussion on limitations and future directions. These improvements have strengthened the scientific impact, clarity and transparency of the work.

Notably, the following aspects of the revision are particularly worthy:

Good new data on NSm–STAT2 interaction and ubiquitin-mediated degradation, including MG132 restoration experiments and co-IP assays.Use of confocal microscopy and quantification tools to support claims of subcellular localisation and trafficking of STAT2.Useful addition of bioinformatic motif analysis to support the hypothesis of E3 ligase recruitment, including conservation of these motifs across CCHFV strains. Usefully this will help guide future work on CCHFV NSm function and its role in immune evasion.Expanded discussion on species-specificity of STAT2 targeting including useful reference to work on SFTSV and STAS2 which brings valuable context and relevance to CCHFV pathogenesis in vivo.Inclusion of appropriate controls and figure clarifications, including statistical annotations and improved legends. 

I believe the study represents a substantive and novel advance in our understanding of how CCHFV evades the host interferon response and support its publication.

Nevertheless, a few minor textual and technical clarifications remain. These are detailed in the reviews comments that follow and should be addressed in your submission.

Response to Reviewers
Revised Manuscript with Track Changes
Manuscript

Shaden Kamhawi

co-Editor-in-Chief

Paul Brindley

co-Editor-in-Chief

**Journal Requirements:**

1) We note that the Data Availability Statement in the online submission form is currently as follows: 'We accept the journal policies on data availability."  Please provide a complete Data Availability Statement in the submission form, ensuring you include all necessary access information. If your research concerns only data provided within your submission, please write "All data are in the manuscript and/or supporting information files" as your Data Availability Statement.

**Reviewers' comments:**

**Key Review Criteria Required for Acceptance?**

**Methods**

-Are the objectives of the study clearly articulated with a clear testable hypothesis stated?

-Is the study design appropriate to address the stated objectives?

-Is the population clearly described and appropriate for the hypothesis being tested?

-Is the sample size sufficient to ensure adequate power to address the hypothesis being tested?

-Were correct statistical analysis used to support conclusions?

-Are there concerns about ethical or regulatory requirements being met?

Reviewer #1: The additional data requested were appropriately conducted and presented in this revised manuscript. The study design is appropriate, as well.

Reviewer #2: All good. Addition of manufacturer reference for all chemical and reagents used in the study will help for reproducibility.

**Results**

-Does the analysis presented match the analysis plan?

-Are the results clearly and completely presented?

-Are the figures (Tables, Images) of sufficient quality for clarity?

Reviewer #1: The results are presented with sufficient quality for clarity.

Reviewer #2: All my comments were adressed in the text and figures

**Conclusions**

-Are the conclusions supported by the data presented?

-Are the limitations of analysis clearly described?

-Do the authors discuss how these data can be helpful to advance our understanding of the topic under study?

-Is public health relevance addressed?

Reviewer #1: The presented data support the main conclusion. The topic is important for public health and its relevance is addressed.

Reviewer #2: The addition of limitation and future directions improved the discussion.

**Editorial and Data Presentation Modifications?**

Reviewer #1: The authors may comment about the upper band of NSm-Flag detected in Figure 1A, B, D, and Fig 2A.

Reviewer #2: (No Response)

**Summary and General Comments**

Reviewer #1: A previous study reported that CCHFV antagonizes IFN-I signaling in human cell lines, but the mechanism of how the virus inhibits IFN-I signaling is not known. In this manuscript, the authors demonstrated that the non-structural protein NSm, suppresses IFN-I signaling in human cell lines. They also showed that NSm binds to STAT2 and induces its ubiquitination and degradation to suppress IFN-I signaling. The authors responded majority of the previous critiques and modified the manuscript accordingly. The data in this revised manuscript are convincing and appropriate for the conclusion made by the authors.

Reviewer #2: Regarding Comment 3 on NSm expression levels:

I understand your rationale for controlling NSm expression levels by adjusting plasmid amounts. However, I believe including both mock-transfected and NSm-transfected cells would be a better strategy to demonstrate the efficiency of your transfection protocol in HEK293T cells. Could you please ensure that you add the specific amount of NSm plasmid used for transfection in your figure legend?

Minor Comments:

Line 131: Please provide the complete reference for LT-1, especially since you mention following the manufacturer's protocol. Who is the manufacturer?

Line 178: Please state IFA in full at first mention.

Lines 214-215: Could you please clarify how you assess ISRE gene activation in HEK293T cells? Which genes did you use, and what were the upregulation levels? This is unclear from both the text and the figure. I think providing more details about your dual-luciferase assay would help the reader. Additionally, in your figure axis labels, please make it clear that you are measuring luciferase activity.

Line 222: Please state the acronym VSIV in full at first mention.

Figure 1D: You mention that HEK293T are IFN-competent cells, but VSIV infection, which is sensitive to IFN action, is inhibited only after addition of IFN. Is this because you are examining early time points?

Line 299: Please state what TGN stands for at first mention.

Line 303: Do you have a quantitative measure for NSm localization in the TGN? The use of the word "partially" is too vague.

Line 374: Please add a reference to support the following statement: "NSm may have a function that regulates RNA expression in the host."

PLOS authors have the option to publish the peer review history of their article (what does this mean? ). If published, this will include your full peer review and any attached files.

**Do you want your identity to be public for this peer review?** For information about this choice, including consent withdrawal, please see our Privacy Policy .

Reviewer #1: No

Reviewer #2: **Yes: ** Marine J Petit

**Figure resubmission:**

**Reproducibility:** To enhance the reproducibility of your results, we recommend that authors of applicable studies deposit laboratory protocols in protocols.io, where a protocol can be assigned its own identifier (DOI) such that it can be cited independently in the future. Additionally, PLOS ONE offers an option to publish peer-reviewed clinical study protocols. Read more information on sharing protocols at https://plos.org/protocols?utm_medium=editorial-email&utm_source=authorletters&utm_campaign=protocols

---

## [Editor Report · Decision Letter 2]

29 Oct 2025

Dear Professor Yasuda,

We are pleased to inform you that your manuscript 'Crimean-Congo hemorrhagic fever virus NSm protein inhibits the type I interferon signaling by binding to STAT2.' has been provisionally accepted for publication in PLOS Neglected Tropical Diseases.

Best regards,

Roger Hewson

Guest Editor

David Safronetz

Section Editor

Shaden Kamhawi

co-Editor-in-Chief

Paul Brindley

co-Editor-in-Chief

Dear Dr Yasuda

Thank you for submitting the revised version of your manuscript.

I have reviewed the revised manuscript alongside the responses to both reviewers’ comments and I am pleased to confirm that all points raised during the initial round of peer review have been addressed.

In particular, I believe the following updates have strengthened the manuscript significantly:

-Clarification of STAT2 degradation via proteasome pathway, supported by MG132 rescue and co-IP experiments

-Confocal microscopy and revised figure legends, including clearer annotation of luciferase assays and statistical information

-Bioinformatic motif analysis, which reinforces the hypothesis of E3 ligase recruitment and highlights conservation across CCHFV strains

-Expanded discussion on species-specific STAT2 targeting, with reference to SFTSV, providing helpful context for understanding CCHFV host range and immune evasion

-Improved methodology sections, with inclusion of manufacturer details, clarified experimental design and defined descriptions of reporter assays and transfection conditions and

-Helpful clarification of limitations, such as the qualitative description of NSm–TGN localisation and the speculative nature of NSm’s role in RNA regulation.

All minor issues raised by the reviewers have also been resolved in the revised text.

As such, I am happy to recommend that the manuscript is now accepted, subject to a final check by the journal's editorial office.

Congratulations on a strong and well revised study that offers important new insights into the immune evasion strategies of CCHFV.

All the best

Roger Hewson

[Guest Editor]

---

## [Editor Report · Acceptance letter]

Dear Professor Yasuda,

We are delighted to inform you that your manuscript, " 

Crimean-Congo hemorrhagic fever virus NSm protein inhibits the type I interferon signaling by binding to STAT2.," has been formally accepted for publication in PLOS Neglected Tropical Diseases.

Best regards,

Shaden Kamhawi

co-Editor-in-Chief

Paul Brindley

co-Editor-in-Chief
